

# Eukaryotic plankton species diversity and community structure in the Xiao Jiang River (the primary tributary of Upper Yangtze River), Yunnan

XueRong Li[1,*], JiShan Wang[2,3,*], YunRui He[1], XiaoJun Yang[4] and Mo Wang[1]

[1] Key Laboratory for Conserving Wildlife with Small Populations in Yunnan, Southwest Forestry University, Kunming, Yunnan, China
[2] Southwest Survey and Planning Institute of National Forestry and Grassland Administration, Kunming, Yunnan, China
[3] Research Center of Asian Elephant, National Forestry and Grassland Administration, Kunming, Yunnan, China
[4] School of Geography and Ecotourism, Southwest Forestry University, Kunming, Yunnan, China
[*] These authors contributed equally to this work.

Corresponding author
Mo Wang, wangmo139@live.cn

## ABSTRACT

The Xiao Jiang River, as a crucial element of ecological restoration in the upper reaches of the Yangtze River, plays an indispensable role in agricultural water utilization and water ecology within its watersheds. The water quality status of the Xiao Jiang River not only impacts local water-ecological equilibrium and economic benefits but also holds paramount importance for sustaining ecosystem health in the Yangtze River basin. Plankton surveys and environmental physicochemical detection were conducted in the major channel region of the Xiao Jiang River in dry and wet periods in 2022 to better understand the diversity of eukaryotic plankton and its community structure characteristics. Environmental DNA is an emerging method that combines traditional ecology with second-generation sequencing technology. It can detect species from a single sample that are difficult to find by traditional microscopy, making the results of plankton diversity studies more comprehensive. For the first time, environmental DNA was used to investigate eukaryotic plankton in the Xiao Jiang River . The results showed that a total of 881 species of plankton from 592 genera in 17 phyla were observed. During the dry period, 480 species belonging to 384 genera within 17 phyla were detected, while, during the wet period, a total of 805 species belonging to 463 genera within 17 phyla were recorded. The phylum Ciliophora dominated the zooplankton, while the phylum Chlorophyta and Bacillariophyta dominated the phytoplankton. The presence of these dominant species indicate that the water quality conditions in the study area are oligotrophic and mesotrophic. Principal coordinate analysis and difference test showed that the number of plankton ASVs, abundance, species richness, dominating species, and diversity indices differed between the dry and wet periods. Spearman correlation analysis and redundancy analysis (RDA) of relative abundance data with environmental physicochemical factors revealed that water temperature (WT), dissolved oxygen (DO), potential of hydrogenacidity (pH), ammonia nitrogen (NH3-N), total nitrogen (TN), electrical conductivity (EC) and the determination of redox potential (ORP) were the main environmental physicochemical factors impacting the plankton community structure. The results of this study can serve as a provide data reference at the plankton

level for water pollution management in the Xiao Jiang River, and they are extremely important for river ecological restoration and biodiversity recovery in the Yangtze River basin.

## INTRODUCTION

The river ecosystem is one of the most significant ecosystems in nature (*Ni & Liu, 2006*). It serves numerous social functions, including freshwater supply, biodiversity preservation, environmental purification, *etc.* (*Luan & Chen, 2004*). A healthy river ecosystem is sustainable and adaptable, meeting social requirements while preserving ecological structure and function (*Zhao et al., 2020*). However, due to climate change and human activities, such as water and soil loss, eutrophication, and biological invasion, there has been an increase in the loss of freshwater biodiversity and deterioration of ecological functioning in recent decades (*Dudgeon et al., 2006*; *Vörösmarty et al., 2010*; *Feio et al., 2021*). Therefore, a thorough understanding of the biological state of river ecosystems is required, and conducting a river biodiversity survey can assist us in achieving this goal. The investigation of river biodiversity lays a foundation for assessing ecosystem resources and provides a scientific basis for understanding the detailed status of river ecosystems. Additionally, it promotes further restoration and protection efforts for threatened river ecosystems.

Plankton, as an integral component of an aquatic ecosystem, significantly contributes to the structure and function of the food web, material transformation, energy flow, information transfer, and other ecological processes (*Park & Shin, 2007*; *Rubin & Leff, 2007*; *Datta, 2011*). At the same time, plankton dynamics depend on nutrients, environmental factors and the presence of other organisms. Therefore, changes in any of these components will impact plankton diversity and abundance (*Williamson et al., 2011*; *Faggotter, Webster & Burford, 2013*; *Edwards et al., 2016*; *Duan, 2019*). As primary producers, characterized by abundant species and short generation time (*Guevara et al., 2009*; *Sun et al., 2013*), phytoplankton's species composition, community structure and abundance are hypersensitive to environmental changes. This makes them important indicators of the aquatic ecosystems (*Blancher, 1984*; *Stevenson et al., 1991*; *Watson, Mccauley & Downing, 1997*; *Lin et al., 2014*; *Wen et al., 2017*). As primary consumers in aquatic ecosystems, zooplankton plays a key role in nutrient transfer between primary producers (phytoplankton) and higher consumers (fish, shrimps, *etc.*), and it exerts top-down control over energy flow and matter cycling in aquatic ecosystems (*Lonsdale, Cosper & Doall, 1996*; *Griffin, Herzfeld & Hamilton, 2001*; *Zakaria, 2015*; *Bess et al., 2021*). Strong seasonal correlations between the concentrations of total nitrogen and chlorophyll in the Mississippi River provides evidence that "nutrients affect the biological responses of ecosystems (*Lohrenz et al., 2008*). Through the investigation of phytoplankton communities

in the main river sections of Tongling City, it was found that phytoplankton density and species number were significantly correlated with chemical oxygen demand (COD), and environmental factors such as nitrogen, phosphorus and pH were the main driving factors affecting the species distribution of phytoplankton in rivers in this area (*Wang et al., 2013*). The findings of a study on the relationship between plankton and environmental factors in tidal rivers in Luoyuan Bay, China, showed that diatoms were negatively correlated with nutrient concentrations, whereas green and cyanobacteria did not show significant correlations with environmental factors (*Pan et al., 2017*).

Environmental factors influencing phytoplankton community structure, such as species composition, cell abundance and species diversity, are not identical under different conditions, and the main drivers vary slightly (*Rhee, 1982*). Several studies have demonstrated that nitrogen, phosphorus or a synergistic effect of both can control phytoplankton growth (*Elmgren & Larsson, 2001*; *Smith, 2003*), biomass (*Cloern, 2001*; *Bledsoe et al., 2004*), and species composition (*Duarte, Agustí & Agawin, 2000*; *Smayda & Reynolds, 2001*). Increased concentrations of nitrogen and phosphorus can enhance the growth of phytoplankton, which indirectly impacting zooplankton density and biomass by supplying food resources (*Li, Yu & Ma, 2014*). By understanding the distribution characteristics of plankton diversity distribution, we can compensate for the lack of water quality evaluation based on physical and chemical indicators, which is crucial for river eutrophication control (*Stevenson et al., 1991*; *Lin et al., 2014*; *Chen et al., 2022*).

Despite the biological importance of plankton in freshwater ecosystems, traditional taxonomic methods used to study plankton communities are unable to achieve large-scale biodiversity surveys due to their time-consuming and labor-intensive nature, a lack of resources and funding species identification, as well as a shortage of experienced taxonomists (*Trebitz et al., 2017*; *Zhao et al., 2021*). With the continuous advancement of sequencing technology, environmental DNA is an emerging detection method that combines traditional ecology with second-generation sequencing technology to survey and assess biodiversity (*Ruppert, Kline & Rahman, 2019*). It can supplement traditional survey methods and to some extent, overcome the limitation that traditional identification methods rely solely on personal experience and ability for identification (*Chain et al., 2016*; *Goldberg et al., 2016*; *Deiner et al., 2017*). The environmental DNA approach was initially used in the study of environmental microbiology, then it was widely recognized and applied in various other study area since 2000 (*Hänfling et al., 2016*). In recent years, environmental DNA has played an irreplaceable role in the research of community structure and diversity of soil fungi, air pollen, fish, amphibians, reptiles, planktonic, and benthic communities (*Buee et al., 2009*; *Kelly et al., 2014*; *Kraaijeveld et al., 2014*; *Thomsen & Eske, 2015*; *Yang et al., 2016*; *Jin et al., 2022*; *Wang et al., 2022*). Due to its ability to achieve rapid and efficient species identification, it has progressively become a powerful tool for large-scale biodiversity research (*Fernando, 2002*; *Shokralla et al., 2012*; *Ji et al., 2013*; *Beng et al., 2016*; *Pawlowski et al., 2016*).

The Xiao Jiang River is a first-level tributary of the right bank of the upper Yangtze River. It is an area in the Yangtze River basin where soil erosion is more serious and the ecological environment is fragile (*Institute of Geographic Sciences and Natural Resources*

*Research, CAS, 1987*; *Chen, You & Zhu, 2000*; *Zhang & Wang, 2006*; *Zhu et al., 2019*). The valley bottom of the mainstream of the Xiao Jiang River's mainline is flat and fertile, with sufficient light and thermal resources. Thus, it has become the primary region for agricultural production (*Liu et al., 2022*). Therefore, the Xiao Jiang River not only handles agricultural and ecological water consumption in the basin, but also plays an important role in the ecological rehabilitation of the upper Yangtze River. Its water quality state affects local economic benefits and is critical to maintaining the health of the Yangtze River basin ecosystem.

The plankton community structure varies to some extent in response to changes in physicochemical factors and nutrients in the water environment. Therefore, it is crucial to understand the composition of plankton species, the dominant species and the characteristics of diversity distribution, in order to effectively assess water quality, and ensure river health control, compensating for the limitationsof physical and chemical indicators. However, no relevant studies on plankton have been carried out in the region previously. Therefore, this study employs an environmental DNA approach to investigate plankton in the Xiao Jiang River for the first time, aiming to comprehend the characteristic differences in plankton diversity during dry and wet periods and unveil the environmental factors that drive changes in plankton community structure. Thise study can provide data reference for managing of water pollution and ensuring the health of water ecology in the Xiao Jiang River basin, which is crucial for river ecological restoration and biodiversity recovery in the Yangtze River basin.

## MATERIALS & METHODS

### Sampling site setup and sample collection

The sampling sites in this study were selected based on the Specifications for Freshwater Plankton Surveys (*Ministry of Agriculture and Rural Affairs of the People's Republic of China, Yangtze River Fisheries Research Institute of Chinese Academy of Fishery Sciences, 2010*) and the Standards for the Investigation of Reservoir Fishery Resources (*Ministry of Water Resources of the People's Republic of China, Institute of Hydroecology, MWR & CAS, 2014*), taking into account factors such as upstream, midstream and downstream locations along rivers, tributary confluence, and human activities interference. Among them, eight sampling sites were set in the Xiao Jiang reach, seven in the Dabaihe reach, and five sampling sites in the Kuaihe reach (Fig. 1, Table 1). Geographical coordinates of the sampling site were collected using a global positioning system (GPS) (Garmin Legend; Garmin USA, Olathe, KS, USA). A total of 120 environmental DNA samples and 40 water samples were collected in dry period and wet period. Three environmental DNA samples were collected at each sampling site using sterilized wide-mouth bottles with a capacity of 1 L. All samples were stored away from light and vacuum filtered within 24 h using 0.22 μm mixed cellulose filter membranes (MCE). The glass devices for filtration were cleaned and sterilized before the start of each sample extraction. The membranes were placed in 5 ml sterilized lyophilization tubes and stored frozen using liquid nitrogen to be transported back to the laboratory.

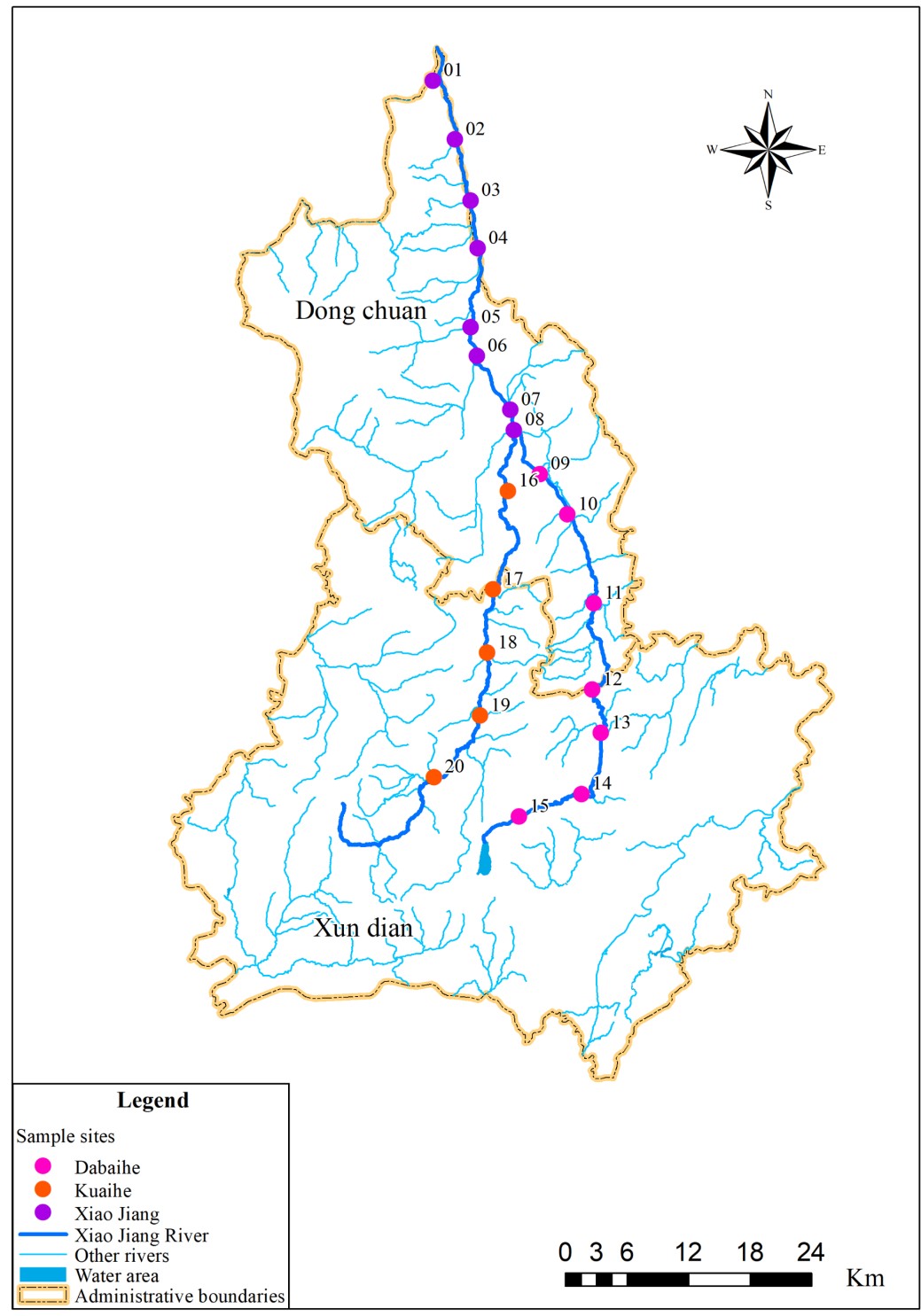

**Figure 1 Sampling distribution map in the Xiao Jiang River.** Vectorized hand mapping of the Xiao Jiang River and water area based on 2022 satellite imagery using ArcGIS software. (continued on next page...)

**Figure 1 (...continued)**
Other rivers were extracted based on DEM (https://www.gscloud.cn) using ArcGIS software for water system analysis. Administrative boundaries by bigemap (http://www.bigemap.com/reader/download/detail201802015.html). The DEM data set is available at Geospatial Data Cloud site, Computer Network Information Center, Chinese Academy of Sciences (http://www.gscloud.cn).

**Table 1** Information of sampling points in the Xiao Jiang River.

| Sampling site | Longitude (N) | Latitude (E) | Altitude (m) |
|---|---|---|---|
| XJ01 | 103°03′04.40″ | 26°30′49.16″ | 795 |
| XJ02 | 103°04′45.71″ | 26°26′14.98″ | 821 |
| XJ03 | 103°05′49.28″ | 26°22′07.56″ | 906 |
| XJ04 | 103°06′19.79″ | 26°18′44.20″ | 979 |
| XJ05 | 103°05′43.71″ | 26°14′30.08″ | 1,078 |
| XJ06 | 103°06′14.63″ | 26°11′15.09″ | 1,096 |
| XJ07 | 103°08′59.05″ | 26°07′14.76″ | 1,126 |
| XJ08 | 103°09′02.21″ | 26°06′13.66″ | 1,143 |
| DBH09 | 103°10′57.35″ | 26°02′56.86″ | 1,225 |
| DBH10 | 103°15′14.06″ | 25°55′31.02″ | 1,450 |
| DBH11 | 103°13′11.38″ | 26°00′19.46″ | 1,352 |
| DBH12 | 103°08′10.98″ | 26°02′43.10″ | 1,207 |
| DBH13 | 103°15′04.85″ | 25°47′59.82″ | 1,828 |
| DBH14 | 103°15′30.89″ | 25°44′44.87″ | 1,863 |
| DBH15 | 103°14′02.11″ | 25°40′28.65″ | 1,995 |
| KH16 | 103°09′07.66″ | 25°38′57.02″ | 2,103 |
| KH17 | 103°07′20.50″ | 25°54′35.83″ | 1,500 |
| KH18 | 103°06′51.18″ | 25°49′53.16″ | 1,632 |
| KH19 | 103°06′14.50″ | 25°47′10.46″ | 1,677 |
| KH20 | 103°02′17.73″ | 25°41′32.09″ | 1,882 |

Water temperature (WT) and dissolved oxygen (DO) were assessed in the field at each sample site using a dissolved oxygen meter while collecting planktonic environmental DNA samples. The 1L water sample was collected using a water collector, sealed in a plastic bottle, stored at a low temperature away from light, and transported back to the laboratory for the determination of other environmental physicochemical factors. The determinations of redox potential (ORP), electrical conductivity (EC), potential of hydrogen (pH), total nitrogen (TN), total phosphorus (TP), ammonia nitrogen (NH3-N), nitrate (NO3-N), phosphate ($PO_4^3$-P), chemical oxygen demand (CODCr), and chlorophyll a (Chl a) concentrations were performed according to the Water and Wastewater Monitoring and Analysis Methods:4th edition (*Edtorial Board of Water and Wastewater Montoring and Analysis Methods, Ministry of Environmental Protection of the People's Republic of China, 2002*).

## DNA extraction and PCR amplification
The total DNA of the plankton was extracted using the Water DNA Kit (Omega, Norcross, GA, USA) and following the manufacturer's instructions. The DNA extraction process was

set up with ddH$_2$O as a negative control. The three biological replicates of each sampling site were pooled to yield more DNA. DNA quality was assesed using a fluorescence quantification instrument (Qubit3.0; Invitrogen) and 2% agarose gel electrophoresis. The V9 region of the nuclear small subunit ribosomal RNA (18S rRNA) was amplified by using 1380F and 1510R PCR primers (*Liu et al., 2017*). PCR was performed with 30 μL reactions for each sample, including 15 μL of PCR Master Mix, 1 μL of forward primer, 1 μL of reverse primer, 3 μL of DNA template, 10 μL of ddH$_2$O. The PCR reaction procedure was performed as follows. 94 °C for 3min, 5 cycles at 94 °C for 30s, 45 °C for 20s, 65 °C for 30s; 20 cycles at 94 °C for 20s, 55 °C for 20s, 72 °C for 30s extension. For each sample, three replicates of the PCR technique were performed, and ddH$_2$O was used as a negative control throughout the experiment. Then, the PCR products of the same sample (three PCR replicates per sample) were mixed and subjected to 2% agarose gel electrophoresis. The mixed product was purified and recovered using Hieff NGSTM DNA Selection Beads. After purification, the qualified PCR products were standardized and subjected to high-throughput sequencing on the Illumina MiSeq platform.

## High-throughput sequencing (HTS) and data processing

All raw datas has been submitted at NCBI, the accession number of NCBI SRR23727913–SRR23727951. Initial quality control was completed using PRINSEQ, setting a 10 bp window size and trimming bases with quality scores below 20 (*Schmieder & Edwards, 2011*). For Cutadapt, adapter overlap threshold for the trimming of 5 bases was used, with an adapter error rate of 0.1 (*Martin, 2011*). The merging of raw paired-end reads was conducted used PEAR (*Zhang et al., 2014*). Sequencing data were analyzed used DADA2 v1.14.0 to generate ASVs tables (adjusted options: truncQ, 2; maxN, 0; maxEE, 2,2). Taxonomy assignment was performed based on the SILVA reference database version 138.1. (*Quast et al., 2013*). For the purpose of this study, the annotations were completed with reference to planktonic taxonomic identification books (*Shen, 1999*; *Witty, 2004*; *Hu & Wei, 2006*; *Dang et al., 2015*; *Bellinger & Sigee, 2015*) Insects, bacteria, fungi, *etc.* were removed from the annotation table, and only phytoplankton and zooplankton were retained (*Garcia-Vazquez et al., 2021*; *Zhao et al., 2021*).

Based on the dominance index calculation method, species with dominance index greater than 0.02 were selected as dominant species (*Dufrêne & Legendre, 1997*). All statistical analyses were done in R 4.0.5. The "ggvenn" package produces a Venn diagram. Showing the species richness and dominant species richness in dry (blue) and wet (yellow) periods. Alpha diversity was characterized by the Shannon, Simpson and Pielou indices, which were plotted on bar plotting. Box charts were used to characterize the dry and wet period changes in the abundance of each plankton phyla. Differences in physicochemical factors in the water column during dry and wet periods depicted by box charts. In this process, the data were tested for normality. Data obeys a normal distribution were used a t.test, and data does not obey a normal distribution were used Wilcoxon.test. Determine whether changes in dry and wet period were significantly different. We have calculated the Bray–Curtis distance for each sample and performed a principal coordinate analysis (PCoA) based on this distance, colouring the samples as from the dry and the wet period. PCoA analysis

was used the "vegan" package. To investigate the water physicochemical factors that drive dry and wet period changes in the plankton community, the abundance data matrix was related to the water physicochemical factor matrix by using the Mantel test, and abundance data were combined with water physicochemical factors for redundancy analysis (RDA). Mantel test analysis of physicochemical factors and abundance in the water column was used the "linkET" package. Redundancy analysis was used the "vegan" package.

# RESULTS

## Results of amplicon sequencing

The sample DBH14dry failed the PCR amplification experiment due to low DNA extraction concentration, and 39 samples (19 in the dry period and 20 in the wet period) were sequenced based on 18S-V9. After quality control, the total number of valid sequences obtained was 4540,657 (including plankton and other eukaryotes). The effective number of sequences for each sample ranged from 83,310 to 149,057 with an average of 116,427. After filtering out other eukaryotes, a total number of 2,398 planktonic ASVs were obtained (Table S1).

## Eukaryotic plankton communities structural features
### Plankton community composition

In this study, a total of 881 species of plankton form 592 genera in 17 phyla were observed. During the dry period, 480 species belonging to 384 genera within17 phyla were detected, while, during the wet period, a total of 805 species belonging to 463 genera within 17 phyla were recorded.

A total of 398 zooplankton species were annotated during the dry and wet periods, with species occurring only in the dry period accounting for 7.5% of the total number of species, species occurring only in the wet period accounting for 45.7% of the total number of species, and species common to both the dry and wet periods accounting for 46.7% of the total number of species (Fig. 2A). There were seven dominant zooplankton species, two in dry periods and six in wet periods (Table 2). One dominant species, *Paramecium multimicronucleatum*, was common to both the dry and wet periods. (Fig. 2B). Zooplankton abundance was 188,006 and the number of ASVs was 1216. The percentage of abundance was 40.54% and 59.46%, and the percentage of number of ASVs was 31.17% and 68.83%, in dry and wet periods (Fig. 2C). Abundance of *Arthropoda and Sarcomastigophora* was higher in dry than in wet periods, and in the other phyla it was higher in wet than in dry periods (Fig. 2D). Zooplankton were dominated by Ciliophora in both dry and wet periods, and the percentage was higher in the wet period than in the dry period (Fig. 2E). In summary, the number of zooplankton species, dominant species, ASVs and abundance differed between the dry and wet periods.

A total of 483 phytoplankton species were annotated during the dry and wet periods, with species occurring only in the dry period accounting for 9.5% of the total number of species, species occurring only in the wet period accounting for 45.3% of the total number of species, and species common to both the dry and wet periods accounting for 45.1% of the total number of species (Fig. 3A). There were 13 dominant phytoplankton species,

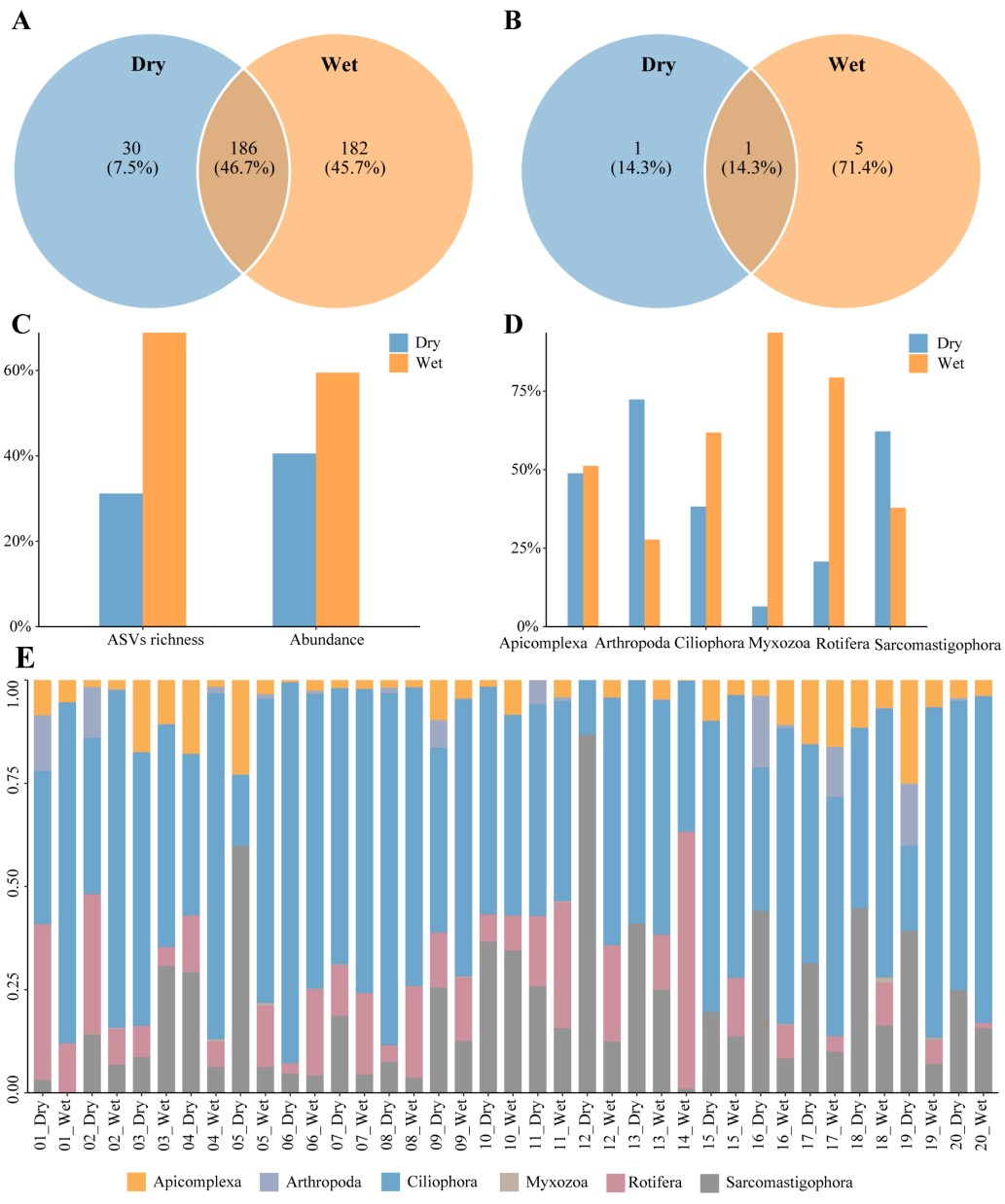

**Figure 2** **Characteristics of zooplankton community.** (A) Species richness, (B) richness of dominant species, (C) proportion of the dry and wet periods, (D) proportion of different phyla in the dry and wet periods, (E) abundance of all samples in the dry and wet periods at the phyla level.

with 8 species in both dry and wet periods (Table 2). Three dominant species, *Cyclotella meneghiniana*, *Cryptomonas pyrenoidifera*, and *Thalassiosira tenera*, were common to both dry and wet periods (Fig. 3B). Phytoplankton abundance was 967,530 and the number of ASVs was 1,652. The percentage of abundance was 32.63% and 67.37%, and the percentage of number of ASVs was 34.93% and 65.07%, in the dry and wet periods (Fig. 3C). Phytoplankton abundance in all phyla was higher in the wet period than in the dry period

**Table 2 Distribution of dominant plankton species.**

| Systematic category | Species | Dry | Wet |
|---|---|---|---|
| Ciliophora | *Bromeliophrya brasiliensis* | | 0.067 |
| | *Cyclidium glaucoma* | | 0.022 |
| | *Paramecium multimicronucleatum* | 0.031 | 0.056 |
| | *Tintinnidium primitivum* | | 0.052 |
| Rotifera | *Ascomorpha ovalis* | | 0.034 |
| | *Synchaeta pectinata* | | 0.023 |
| Sarcomastigophora | *Neobodo designis* | 0.059 | |
| Bacillariophyta | *Cocconeis pediculus* | 0.022 | |
| | *Cyclotella meneghiniana* | 0.125 | 0.088 |
| | *Navicula cari* | 0.023 | |
| | *Thalassiosira tenera* | 0.039 | 0.054 |
| Chlorophyta | *Cladophora vagabunda* | 0.065 | |
| | *Desmodesmus communis* | 0.037 | |
| | *Golenkinia brevispicula* | | 0.024 |
| | *Hafniomonas montana* | | 0.038 |
| | *Hafniomonas reticulata* | | 0.023 |
| | *Hydrodictyon reticulatum* | 0.034 | |
| | *Wislouchiella planctonica* | | 0.030 |
| Cryptophyta | *Cryptomonas pyrenoidifera* | 0.045 | 0.114 |
| | *Teleaulax amphioxeia* | | 0.032 |

(Fig. 3D). Phytoplankton were dominated by Bacillariophyta and Chlorophyta in both the dry and wet periods, with Bacillariophyta being more predominant in dry period than in wet period, and Chlorophyta being more predominant in the wet period than in the dry period (Fig. 3E). In summary, the number of phytoplankton species, dominant species, ASVs and abundance differed between the dry and wet periods.

### Changes in plankton diversity

The zooplankton Shannon, Simpson and Pielou indices varied in the ranges of 1.13–4.46, 0.35–0.98, and 0.22–0.68 (Fig. 4A). At the species richness, the Shannon and Simpson indices were significantly lower in the dry period than in the wet period, and there were no differences in the Pielou indices (Figs. 5A, 5B, 5C). Select taxa with abundance of zooplankton greater than 2%. Perform differential testing for ASVs richness. Perform differential testing for abundance. The number of ASVs in each phylum was significantly lower in the dry period than in the wet period (Figs. 6A–6D). Apicomplexa and Ciliophora abundance did not differ significantly between the dry and the wet periods (Figs. 6E, 6F). Rotifera abundance was significantly lower in the dry period than in the wet period (Fig. 6G). Sarcomastigophora abundance was significantly higher in the dry period than in the wet period (Fig. 6H).

The phytoplankton Shannon, Simpson and Pielou indices varied in the ranges of 0.81–4.52, 0.43–0.98, and 0.29–0.62 (Fig. 4B). At the species richness, the Shannon and Simpson indices were significantly lower in the dry period than in the wet period, and there were

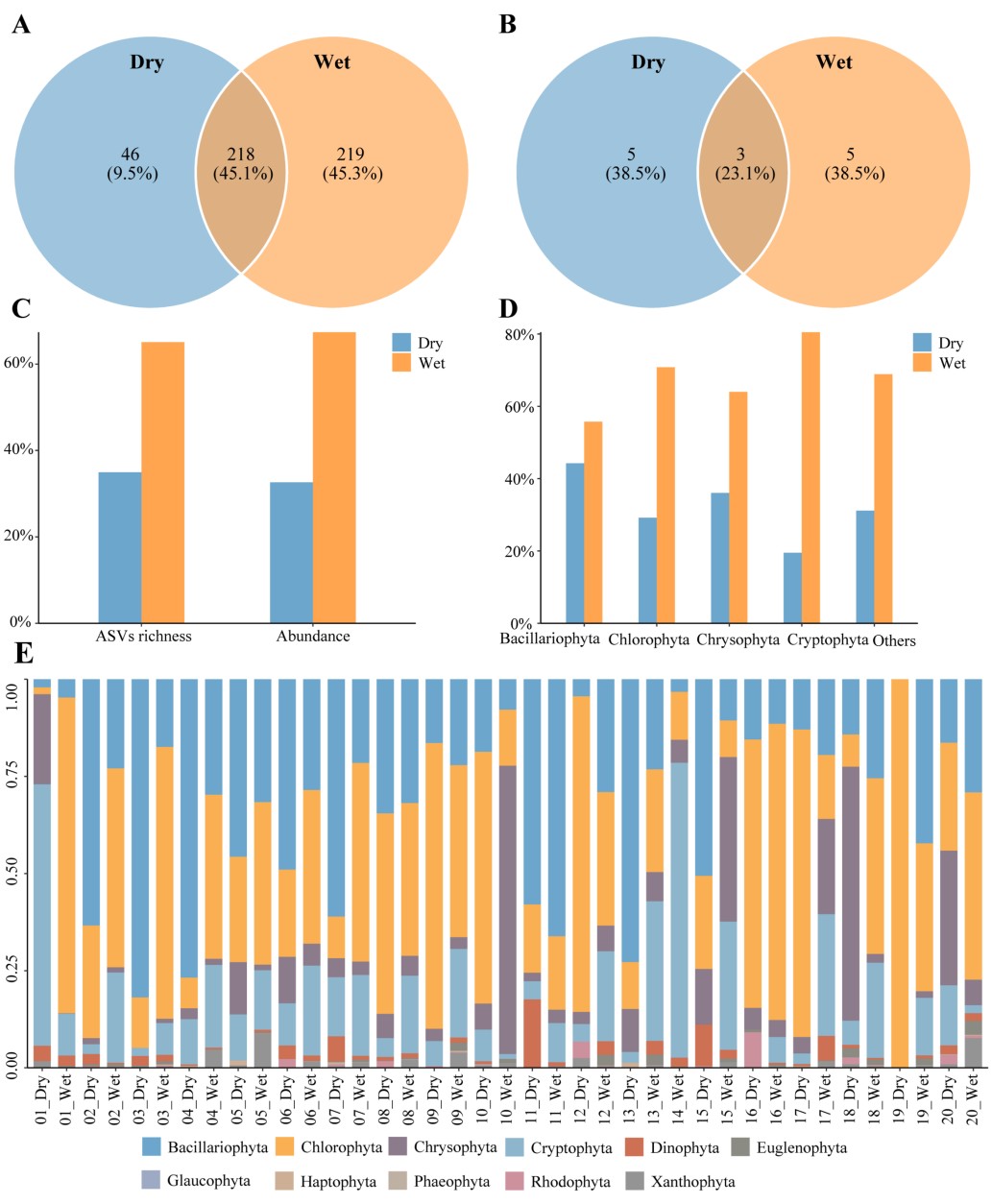

**Figure 3 Characteristics of Phytoplankton community.** (A) Species richness, (B) richness of dominant species, (C) proportion of the dry and wet periods, (D) proportion of different phyla in the dry and wet periods, (E) abundance of all samples in the dry and wet periods at the phyla level.

no differences in the Pielou indices (Figs. 5D, 5E, 5F). Select taxa with relative abundance of phytoplankton greater than 2% Perform differential testing for ASVs richness. Perform differential testing for abundance. The number of ASVs in each phylum was significantly lower in the dry period than in the wet period (7A–7D). Bacillariophyta and Chrysophyta abundance did not differ significantly between the dry and the wet periods (Figs. 7E, 7G).

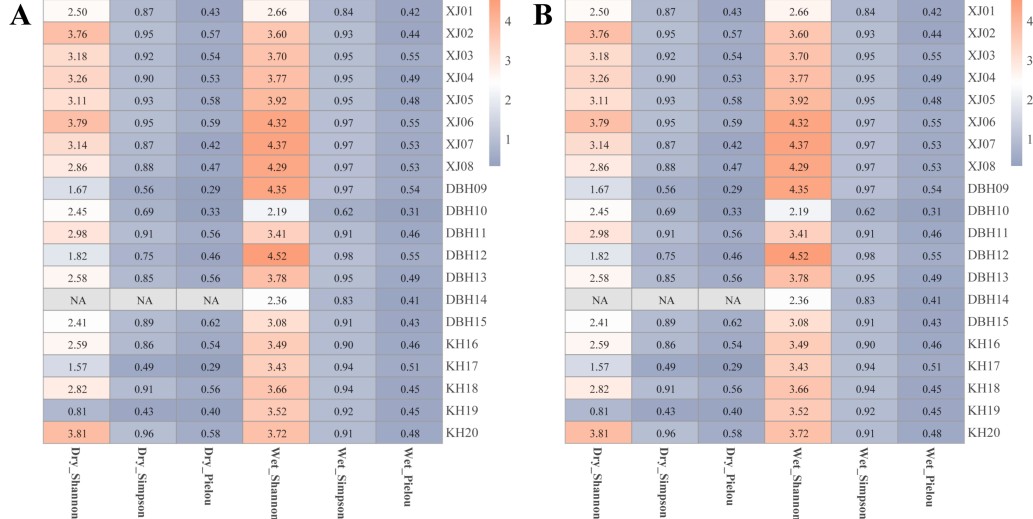

**Figure 4  Alpha diversity of the entire plankton in each sample.** (A) Zooplankton, (B) phytoplankton.

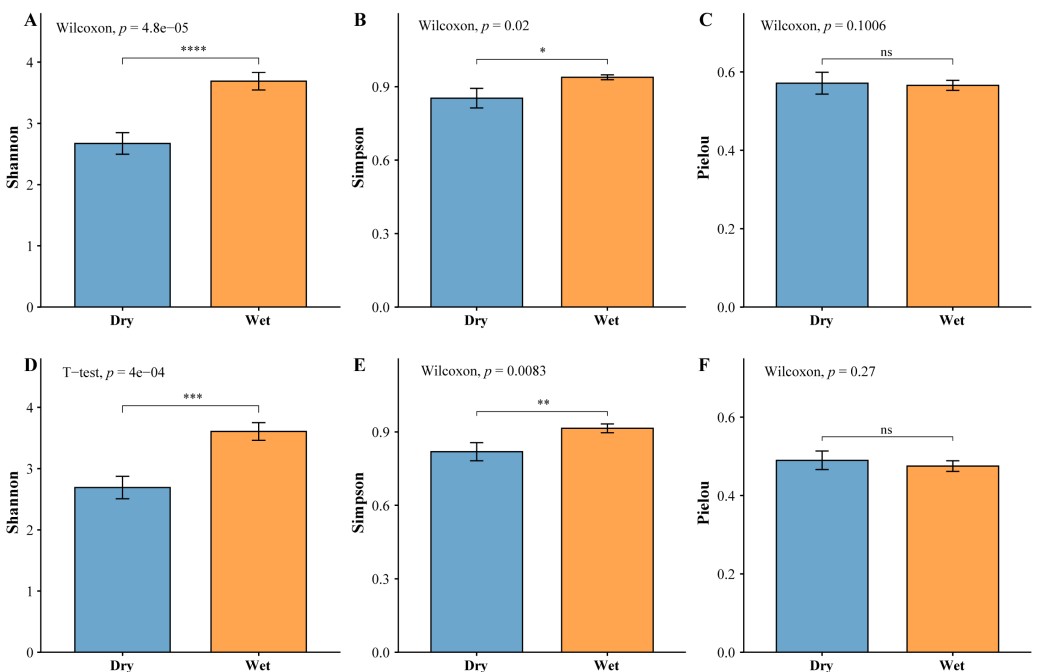

**Figure 5  Test of difference based on the abundance of planktonic diversity.** (A) Zooplankton Shannon, (B) Zooplankton Simpson, (C) Zooplankton Pielou, (D) Phytoplankton Shannon, (E) Phytoplankton Simpson, (F) Phytoplankton Pielou.

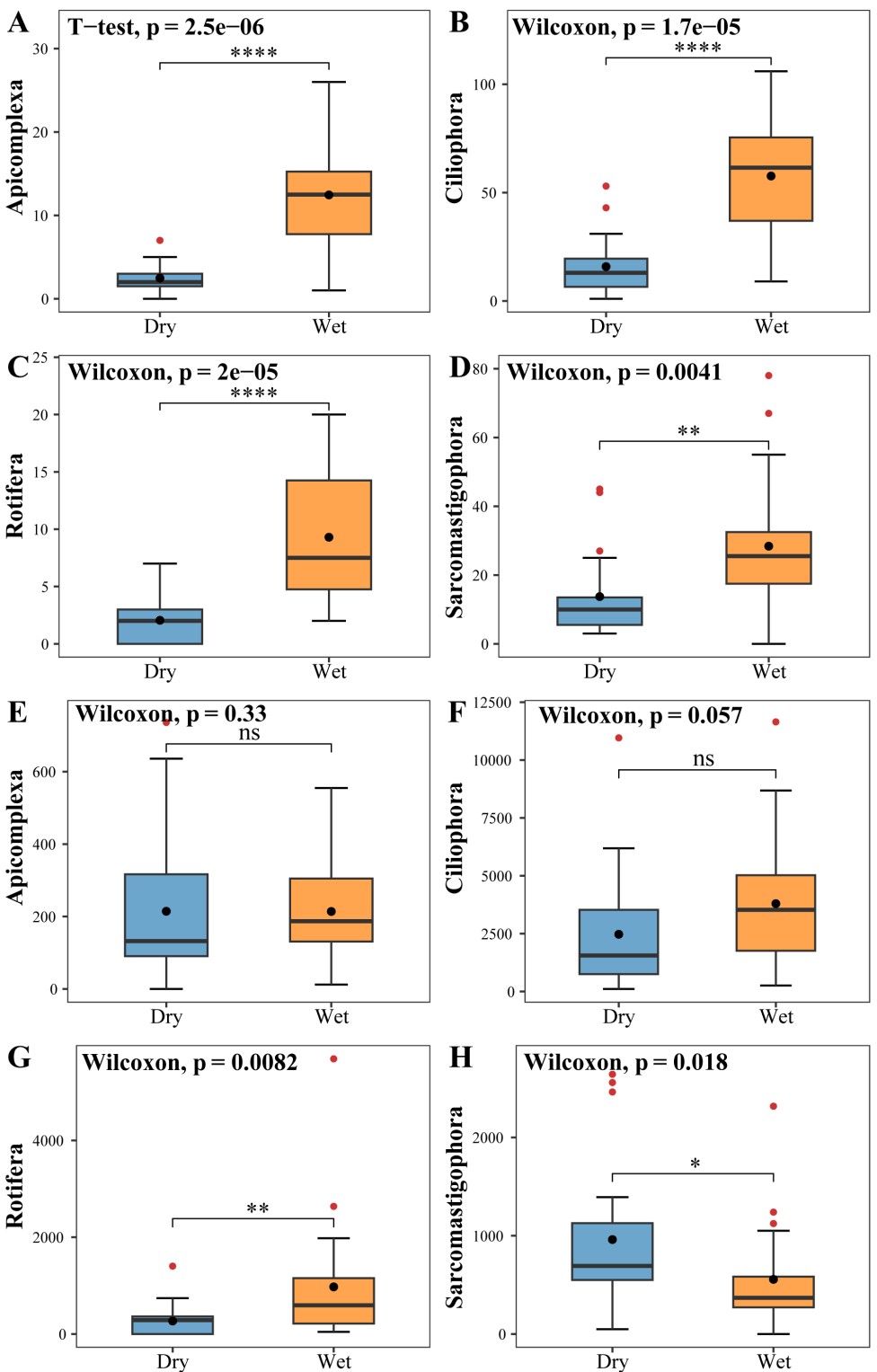

**Figure 6** **Tests for differences in the number of zooplankton (A, B, C, D) ASVs and abundance (E, F, G, H).** The vertical axis of A, B, C & D indicates the number of ASVs in each phylum; E, F, G & H vertical axis represents the abundance of each phylum.

Chlorophyta and Cryptophyta abundance was significantly lower in the dry period than in the wet period (Figs. 7F, 7H).

PCoA analyses were performed on zooplankton and phytoplankton with confidence intervals set at 90%. The zooplankton and phytoplankton samples from most of the sampling sites were split into two distinct clusters for the dry and wet periods (Fig. 8). Overall, the entire plankton community assemblage showed significant differences between the dry and wet periods (PERMANOVA, $p = 0.001$).

## Relationship between eukaryotic plankton and environmental physicochemical factors

The measured values of physicochemical factors of water bodies in the study area during the dry and wet periods are shown in Table S2. TP, NO3-N and ORP were significantly higher in the wet period than in the dry period. DO, pH, TN and EC were considerably higher in the dry period than in the wet period. The remaining physicochemical factors (WT, CODcr, NH3-N, $PO_4^3$-P, and Chlorophyll a) were marginally different between two periods (Fig. 9).

Relative zooplankton and phytoplankton abundance data were correlated with environmental physicochemical factors by the Mantel test to better understand the environmental factors driving changes in plankton diversity during the dry and wet periods. Plankton and environmental variables were used with the Bray–Curtis distance and Euclidean distance, respectively. The results showed that WT, DO, pH, NH3-N, TN, EC and OPR were the main environmental physicochemical factors driving the changes in plankton abundance (Fig. 10). In addition, the environmental physicochemical factors driving changes in the zooplankton community were not consistent with the phytoplankton. TN and EC had the most effect on zooplankton community changes (Fig. 10A), while WT, DO, pH, NH3-N, and OPR had the greatest influence on phytoplankton community changes (Fig. 10B). In conjunction with the abundance of zooplankton and phytoplankton, the main environmental physicochemical factors in the Mantel test were selected for redundancy analysis (RDA). The results showed that the seven environmental physicochemical factors explained 60.35% and 55.69% of the variation in zooplankton and phytoplankton communities in RDA axis 1, respectively. RDA axis 2 explained 24.13% and 29.55% of the variation in zooplankton and phytoplankton communities, respectively (Fig. 11).

## DISCUSSION

### Structural characteristics of eukaryotic plankton communities

The hydrological characteristics and trophic status of rivers tend to change in response to seasonal changes in river flows, which in turn have an impact on plankton community composition (*Biggs & Smith, 2002*; *Thomaz, Bini & Bozelli, 2007*). Because of the seasonal turnover of abiotic (*e.g.*, water temperature) and biotic factors, the plankton community also exhibits seasonal patterns of change (*Zhang et al., 2019*; *Wang et al., 2020*). The results showed that there were substantial variations in plankton species composition, abundance, dominant species, and diversity indices between the dry and wet periods. Zooplankton are dominated by minitype individuals of Ciliophora, Sarcomastigophora and Rotifera,

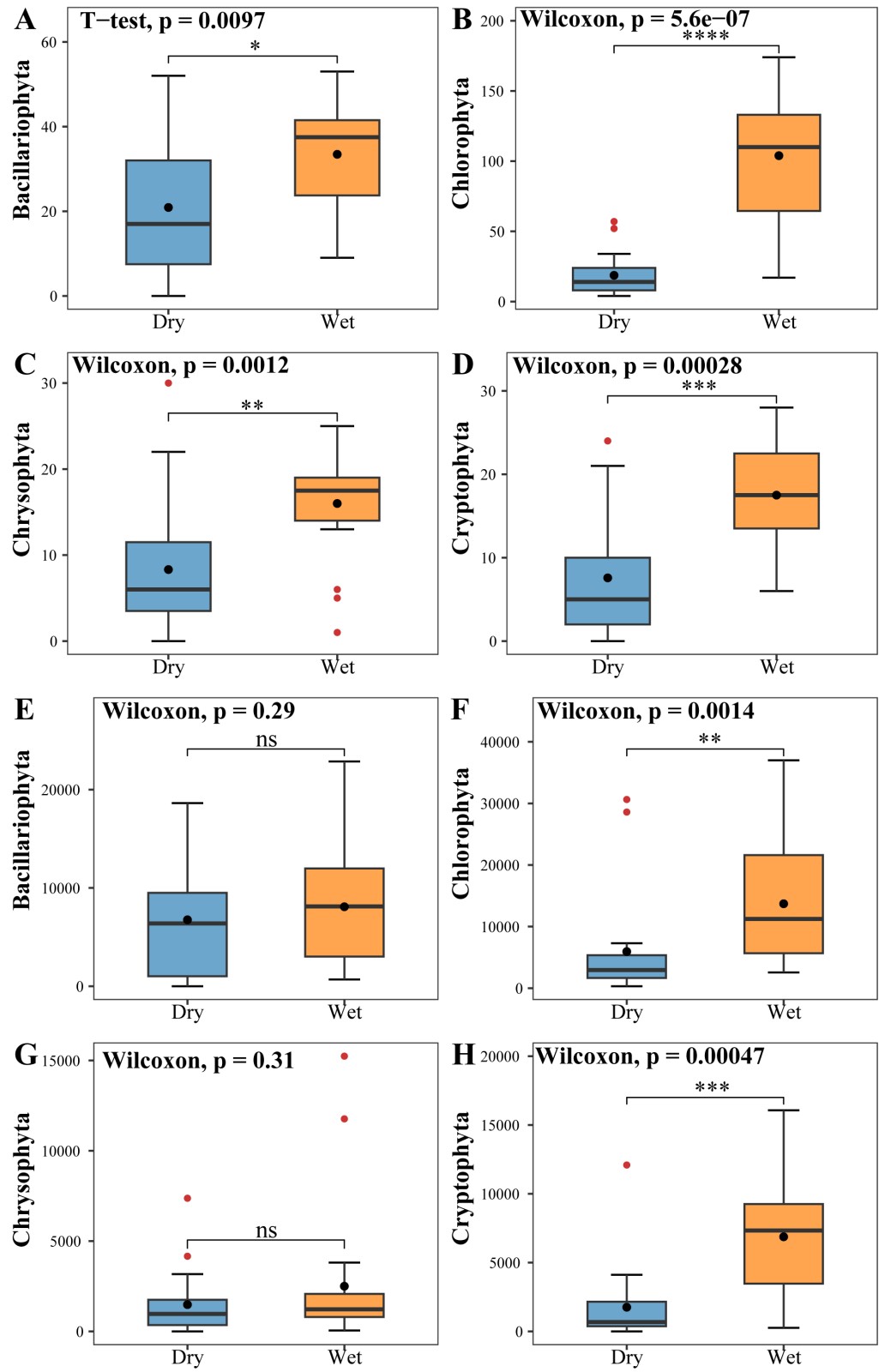

**Figure 7** **Tests for differences in the number of phytoplankton (A, B, C, D) ASVs and abundance (E, F, G, H).** The vertical axis of A, B, C & D indicates the number of ASVs in each phylum; E, F, G & H vertical axis represents the abundance of each phylum.

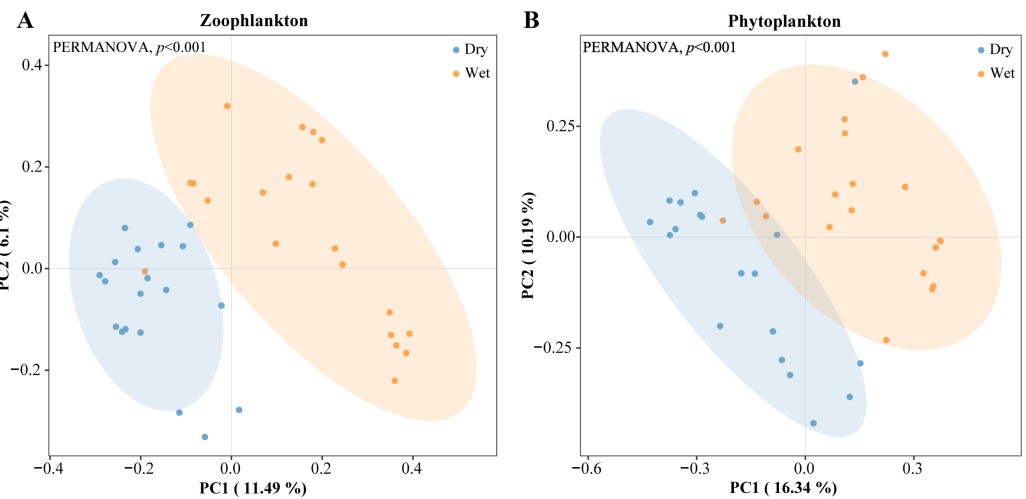

**Figure 8  Clustering of the plankton by dry and wet period using principal coordinate analysis (PCoA) based on Bray–Curtis dissimilarity index.**

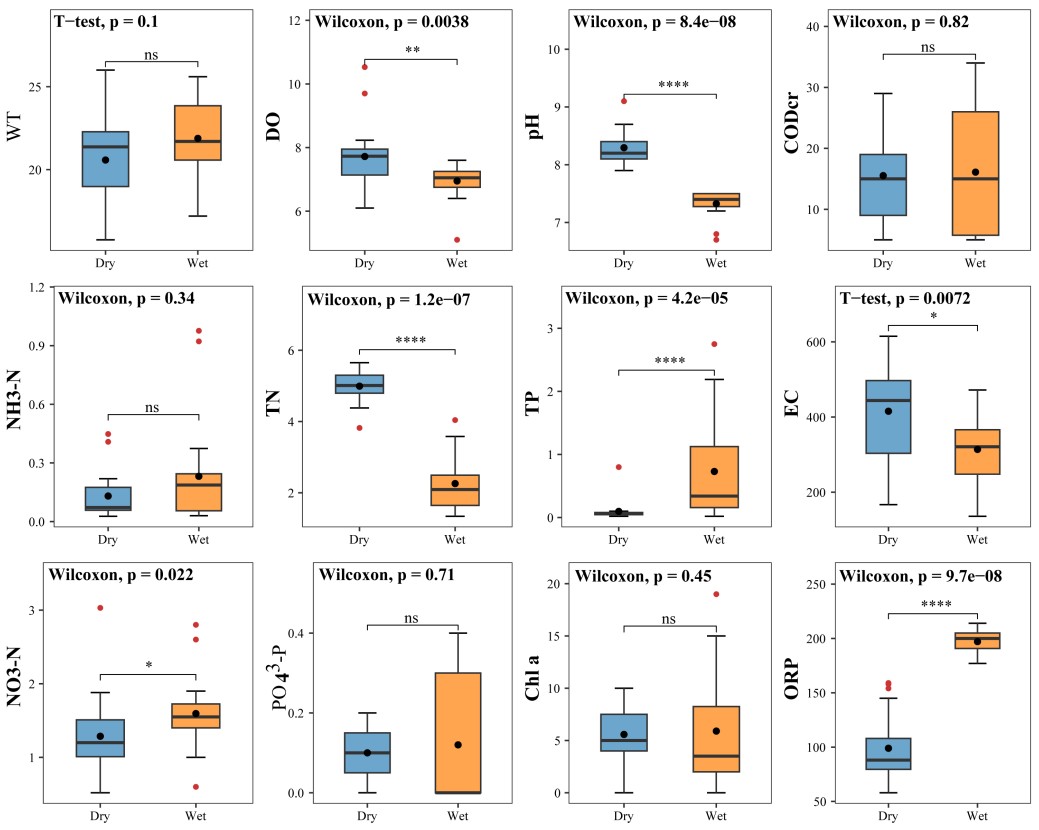

**Figure 9  Difference test for physical and chemical factors in water bodies.**

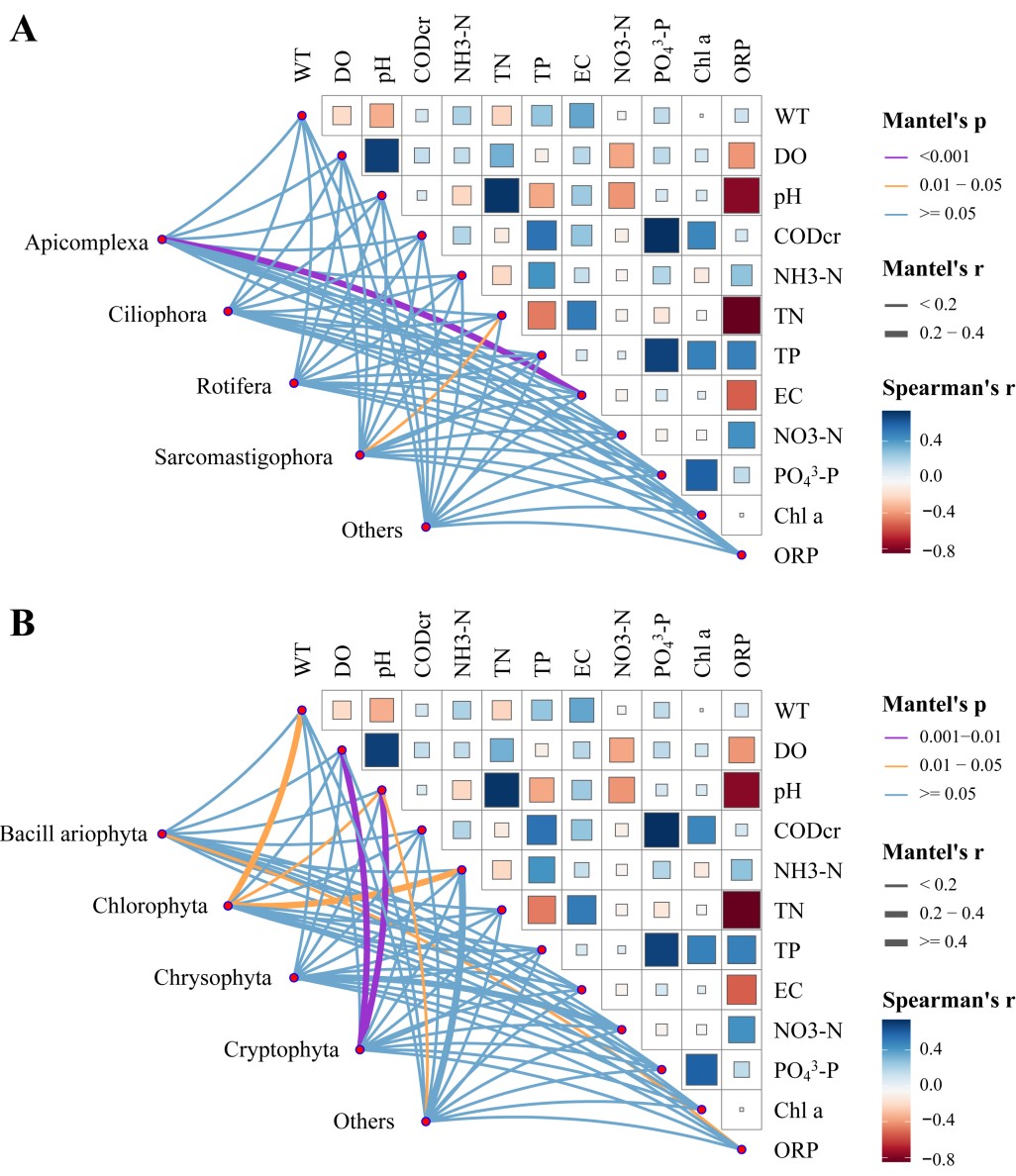

**Figure 10** **Water environmental factors driving dry and wet period changes in the plankton Pairwise comparison of water environmental factors was achieved by using Spearman tests, and the color gradient represents Spearman's correlation coefficient.** (A) Zooplankton, (B) phytoplankton.

which accounted for 94.09% of the total zooplankton abundance. Minitype individual zooplankton were dominated by Ciliophora, which accounted for 65.39% of the total zooplankton abundance. The large zooplankton copepods and branchiopods were less abundant, with their abundance accounting for only 1.32% of the total zooplankton abundance. Phytoplankton was dominated by Bacillariophyta and Chlorophyta, which accounted for 69.99% of the total phytoplankton abundance. The result is similar to the community composition of plankton in other rivers (*Wu, Li & Chen, 2015*; *Chen et al., 2019*; *Chen et al., 2022*; *Yang et al., 2022*; *Xu et al., 2022*). First, rotifers' superior dispersion

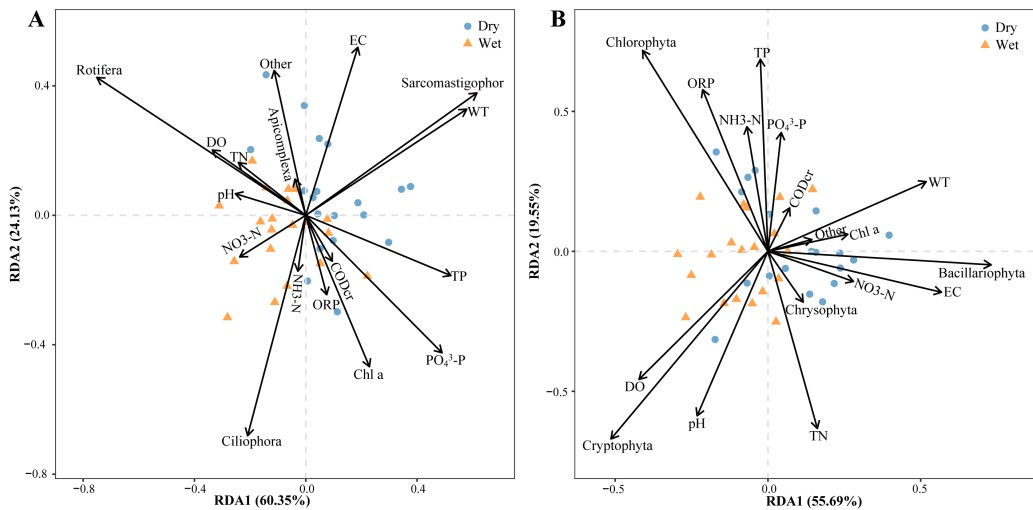

**Figure 11** **Abundance redundancy analysis (RDA) of the abundance of the plankton community.** (A) Zooplankton, (B) phytoplankton.

capacity, parthenogenesis and highly adaptable masticatory apparatus allow them to generate a community from a single dormant individual (*Segers, 2008*). Second, when phytoplankton biomass reaches its peak, higher temperatures can accelerate phytoplankton metabolism and promote phytoplankton growth rates (*Reynolds, 2006*), resulting in increased primary production in the region (*Doney, 2006*; *Chavez, Messie & Pennington, 2011*), and thus indirectly increasing zooplankton biomass (*Lewandowska et al., 2014*). Therefore, the temperature may be one potential reasons for the higher relative abundance of plankton and the greater number of species in the wet period compared to the dry period. The optimum temperature for protozoan growth is 10~25 °C (*Shen, 1999*), which may account for the large number of species and abundance of protozoa in both surveys of this study.

Dominant species play a significant role in the community structure (*Qian et al., 2022*). In terms of dominant species composition the dominant zooplankton species include small Ciliophora, Sarcomastigophora and Rotifera. The dominant phytoplankton species consisted of species in Bacillariophyta, Chlorophyta and Cryptophyta. Rotifers are considered as important bioindicator in depicting trophic status of water quality in ecosystem (*Sládeček, 1983*; *Beērziņš & Pejler, 1987*; *Kulkarni & Zade, 2018*). Among the dominant species in this study, those used to indicate the nutrient status of water quality were also found. *Ascomorpha ovalis*, *Filinia longiseta* and *Synchaeta pectinate* are oligotrophic and mesotrophic water body indicator species (*Sládeček, 1983*; *Beērziņš & Pejler, 1987*; *Mnatsakanova, 2016*). *Cyclotella sensu lato* taxa are a group of diatoms that are frequently dominant members of phytoplankton communities in low-productivity, oligotrophic (*Willen, 1991*; *Hörnström et al., 1993*). The indicative role of some of the dominant species suggests that the water quality conditions in the study area are oligotrophic and mesotrophic. Therefore, it is necessary to strengthen the monitoring

of pollution in the basin and strictly control new sources of pollution, so as to provide strong support for the sustainable development of water ecology in the upper Yangtze River basin.

## Relationship between eukaryotic plankton and environmental physicochemical factors

Environmental physicochemical factors can affect the plankton community directly or indirectly. The results of this study showed that zooplankton community changes were mainly influenced by TN and EC, whereas phytoplankton community changes were mainly influenced by WT, DO, pH, NH3-N, and ORP. Water temperatures are highest during wet period when DO is lowest. The trends of DO and pH were generally consistent among the sample sites. TN, TP and NH3-N showed significant differences between the dry and wet periods. The lower TN abundance in the wet period than dry period may be due to the fact that the denitrification rate rises with increasing water temperature, which helps to reduce the nitrogen load to the water column to some extent (*Paerl et al., 2011*). External runoff, atmospheric deposition, and internal sediment release are primary sources of phosphorus (*Reed, Carpenter & Lathrop, 2000*; *Xu et al., 2010*). According to *Lai, Yu & Gui (2006)*, based on the geographical distribution and mechanistic SWAT model, agricultural fertilization contributes 15% cent of total nitrogen and 10% cent of total phosphorus discharged to rivers, and animal husbandry contributes 14% cent of total nitrogen and 11% cent of total phosphorus discharged to rivers. The Xiao Jiang River is a semi-mountainous river of rainfall origin that receives its mostly from surface runoff and the middle-lower reaches of the Manta Valley, which benefits agricultural production (*Chen, You & Zhu, 2000*; *Liu et al., 2022*). Thus, agricultural fertilizers may be the primary cause of the greater phosphorus levels in the wet period to the dry period. The effects of nitrogen, phosphorus or their synergistic effects on phytoplankton dynamics, although moderated by other factors (*i.e.,* light intensity, pH), and changes in their concentrations are the main factors driving phytoplankton growth, species composition and biomass in the region (*Cymbola, Ogdahl & Steinman, 2008*; *Paerl et al., 2011*; *Wang et al., 2013*). Therefore, changes in nitrogen and phosphorus concentrations may be the primary cause for the increase in biomass during the wet period in this study.

## CONCLUSIONS

The structural characteristics of the zooplankton community in the Xiao Jiang River showed significant changes during dry and wet periods. The indicative role of some of the dominant species suggests that the water quality conditions in the study area are oligotrophic and mesotrophic. WT, DO, pH, EC, OPR, TN and NH3-N are important environmental physicochemical factors that affect the changes of plankton communities during dry and wet periods. The results of the study provide data references at the plankton level for biodiversity conservation and river ecological restoration in the Yangtze River Basin.

## ACKNOWLEDGEMENTS

We are grateful to the team of Dr. JunXing Yang (Kunming Institute of Zoology, Chinese Academy of Sciences, KIZ) for his support of this study. We gratefully acknowledge Mr. ShuWei Liu for his optimizing for sample collection. We would like to thank JinMing Bai (Southwest Forestry University, SWFU) for his help in sample collection. We would like to express gratitude to ZhuoYun Jiang (Southwest Forestry University, SWFU) for her help in data analysis. We also appreciate WenDi Ma (Southwest Forestry University, SWFU) for her assistance in improving the linguistic description of the paper.

### Funding

This work was supported by the Natural Science Research Foundation of Yunnan Provincial Department of Education (No. 2021J0180) and the Natural Science Foundation of Yunnan Province of China (No. 202201AT070046). The funders had no role in study design, data collection and analysis, decision to publish, or preparation of the manuscript.

### Grant Disclosures

The following grant information was disclosed by the authors:
Natural Science Research Foundation of Yunnan Provincial Department of Education: No. 2021J018.
Natural Science Foundation of Yunnan Province of China: No. 202201AT070046.

### Competing Interests

The authors declare there are no competing interests.

### Author Contributions

- XueRong Li performed the experiments, analyzed the data, prepared figures and/or tables, and approved the final draft.
- JiShan Wang conceived and designed the experiments, authored or reviewed drafts of the article, and approved the final draft.
- YunRui He performed the experiments, analyzed the data, prepared figures and/or tables, and approved the final draft.
- XiaoJun Yang analyzed the data, authored or reviewed drafts of the article, and approved the final draft.
- Mo Wang conceived and designed the experiments, authored or reviewed drafts of the article, and approved the final draft.

### DNA Deposition

The following information was supplied regarding the deposition of DNA sequences:
The sequences are available at NCBI: PRJNA941381.

## Data Availability

The raw data is available in the Supplementary File and the sequences are available at NCBI: PRJNA941381.

## Supplemental Information

Supplemental information for this article can be found online at http://dx.doi.org/10.7717/peerj.17972#supplemental-information.

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
