# Peer review of "Eukaryotic plankton species diversity and community structure in the Xiao Jiang River (the primary tributary of Upper Yangtze River), Yunnan"

_PeerJ, doi:10.7717/peerj.17972_

## Round 0.1 · original submission · Major Revisions

While the reviewers see merit in your article, they have suggested and recommended many changes you need to implement to improve the quality and clarity of your ms. During the resubmission of your revised manuscript, make sure to submit the raw data and corresponding scripts upon which the paper is based for evaluation of reproducibility.

·

Basic reporting

At first, I would to mention the actual topic of the study – to compare the freshwater microbial community with of different sites of Xiao Jiang River.
Bioinformatics approaches described in the manuscript can be improved. There are strong reasons to use amplicon sequence variants (ASVs) instead of OTUs for less noise and avoid spurious taxons (https://www.mdpi.com/2306-5354/9/4/146).
Callahan, McMurdie and Holmes points that “the improvements in reusability, reproducibility and comprehensiveness are sufficiently great that ASVs should replace OTUs as the standard unit of marker-gene analysis and reporting” Callahan, B., McMurdie, P. & Holmes, S. Exact sequence variants should replace operational taxonomic units in marker-gene data analysis. ISME J 11, 2639–2643 (2017). https://doi.org/10.1038/ismej.2017.119
There are no raw data published, so it’s impossible to reproduce the whole analysis or perform the further meta-analysis. I strongly suggest to submit raw reads to GenBank and add its access numbers to the manuscript.
What values are for sample DBH14 in Table 3?

Experimental design

Some tables will be clearer and easier to understand as heatmap (Table 3) and histogram (Table 4)
In several comparisons authors use the Mantel test, but don’t report the value of estimated “distance” (L285, L293).
The confidence intervals are intersecting (Figure 7A, 7B, 7C, 7D, 7G). It shows that statistically significant difference is questionable and requires accurate statistical evaluation of the data.
Some details of the study must be clarified to better understanding and reproducibility:
L196: how raw data quality filtering (FASTQ) was performed in details?
L197: which version of USEARCH was used?
L199: What protocol and version of SILVA reference database was used for taxonomic analysis?
As said above, using OTUs approach produces a lot of spurious OTUs. L113 shows thousands of OTUs, most of which, I propose, share less than 1%. Such classification is hard to interpret. For further study see, f.e.:
https://doi.org/10.1371/journal.pone.0227434
https://doi.org/10.1038/s43705-021-00033-z

Validity of the findings

The data analysis contains multiple flaws and no raw data limits the reproducibility.
My notes about the main text:
• Abbreviation “Chl a” is misleading
• Some captions contains errors: f.e.“Tab 4” (page 54)
• Replace “Time” to “Season” at Table 4

·

Basic reporting

Use of English

Various flaws on the use of English have been detected throughout the manuscript. The English language should be improved to ensure that an international audience can clearly understand the manuscript. Although it is impossible at this stage of revision to proofread the entire manuscript, some examples where the language could be improved include:

- Abstract:

Lines 21-24: This sentence is difficult to understand. The authors should consider split it into two sentences. Also the authors should remove "as" from the beginning of the sentence.

Lines 33-34: It is not clear what the authors mean with "the dominant species of zooplankton signal the water quality in this study area". Please rephrase this part in a more simple way.

Lines 61-63 : This sentence was difficult to understand at first read. A suggestion on how to rephrase that sentence: “At the same time, plankton dynamics depend on nutrients, environmental factors and the presence of other organisms, thus changes in any of the above components will impact plankton diversity and abundance (Duan, 2019).”

- Introduction:

Lines 113-114: “planktonic, and benthic”
Replace with “plankton, and benthos” or with “planktonic, and benthic communities”

Lines 134-135: “Since the plankton community structure varies in response to changes in physicochemical factors and nutrients in the water environment to some extent.”
The authors should remove “since” from the beginning of the sentence.

Lines 139-140: The authors could consider rephrasing
“However, it is unclear how abundance plankton are in the Xiao Jiang River and how they are influenced by environmental physicochemical factors.”
As
“However, it is unclear how abundant plankton taxa are in the Xiao Jiang River and how they are influenced by environmental physicochemical factors.”

- Materials and Methods:

Line 177 : “following the manufacturer’s the instructions.”
Remove “the” before instructions

Line 179 : “DNA quality was assess using a”
assesed

Lines 223-224: “ After sieving out other eukaryotes, A total number of 1596 planktonic OTUs were obtained (Tab S1).”
There is a capital “a” in the middle of the sentence. Also, it is advisable to replace the verb “sieving” with “filtering”, because it is a more common wording to use for this kind of treatment.

- Results:

Lines 262-265: “The number and relative abundance datasets of OTUs at the taxonomic level of the phytoplankton phylum were tested for the difference to explore the seasonal variation of each plankton taxon, in which phytoplankton was selected for analysis with a relative abundance percentage greater than 5% (Fig 6, Fig 7).”
It is very difficult to understand the message of this sentence. Do the authors intend to describe the results of two figures in one sentence? It is advisable to describe the results of each figure in a separate sentence. Also the phylum taxonomic level is a rather high taxonomic rank to perform these analyses that are generally performed in lower taxonomic ranks.

- Discussion:

Lines 305-307: “The zooplankton was dominated by microprotozoa (Ciliophora, Sarcomastigophora) and rotifers (Rotifera), accounting for 93.01% of the total number of species.The species number (73.26% and 75.89%) and relative abundance (69.30% and 64.62%) of zooplankton species were dependent on protozoa. In comparison, the number of macrozooplankton copepods and cladoceran (Arthropoda) is minimal relatively.”
This sentence was confusing and it would be advisable to be rephrased, as the message is not clear. First, it is not clear what the numbers in parenthesis refer to. After reading the text a couple of times, I suppose that they refer to microprotozoa, but the authors should be more explicit in their description to make reading more effortless. In addition, it is not clear how the zooplankton is “dependent on” protozoa. Does it mainly comprise of protozoa? If this is the message that the authors would like to convey, then they should consider rephrasing that part accordingly. Finally, the use of the phrase “in comparison” is confusing, given that the authors do not directly compare any samples.

Lines 311-313: “First, rotifers’ superior dispersion capacity, parthenogenesis and highly adaptable masticatory apparatus of allow them to generate a community from a single dormant individual (Segers, 2007).”
Remove “of” before “allow them”

Line 321: “Dominant species play a dominant role in formating of community structure (Qian, 2022).”
The phrasing here is somehow redundant. Maybe consider rephrasing as follow:
“Dominant species are decisive for the community structure.”
“Dominant species play a significant role in the community structure.”

Background and Literature

The authors have nicely structured the introduction, starting from the general need to understand rivers’ biodiversity in order to understand the status of river ecosystems, describing how plankton contributes significantly to the structure and function of river ecosystems, stating how environmental variables influence plankton community structure, and finally describing their study system and presenting the necessity of the present study. However, the second paragraph of the introduction is rather extensive including plankton diversity and environmental variables, and could be split in two. Regarding the cited literature, although it is perfectly acceptable to cite sources in another language, the authors may consider to add references written in English wherever it is possible, maintaining of course the Chinese references. This would facilitate international readers to keep up with the literature and would make the research presented more widely contrastable. This is especially important in the results and discussion section where the authors make important claims that need further citation.

For example, in lines 329-333 the authors state:
“According to current studies, it is generally accepted that Bacillariophyta represents the dominant species in oligotrophic water bodies, whereas Chlorophyta is the dominant species in mesotrophic water bodies (Wang et al., 2022). The dominant zooplankton and phytoplankton species in the XiaoJiang River indicated oligotrophic and mesotrophic water quality conditions.”
Although there is an extensive literature in the use of diatoms (Bacillariophyta) as bioindicators, the authors provide only one reference, which is not contrastable by international readers as it is in Chinese, to support their claim that their study system is oligotrophic to mesotrophic. The mere presence of Bacillariophyta in their samples is not enough to assess the trophic state of their study system as there are diatom species such as Melosira varians (present in some of the samples of this study) that are abundant in poor quality water bodies, thus being indicators of eutrophication. The authors should examine their results more carefully and perform a more exhaustive evaluation of the trophic state of their study system using more than one studies from the currently available literature to support their findings.

Article structure, figures, tables. Raw data shared.

The structure of the article conforms to the acceptable format of standard sections. The submission seems to be self-contained, representing an appropriate unit of publication, and including most of the relevant results.

In Figure 2, the authors talk about “dominant species”, but in the text they do not define which are those species. The authors present their results at phylum level. Thus, they should consider add more detailed information in their results section regarding the dominant species. In addition, they should pay extra attention when they refer to taxa, species, or OTUs. It seems that there is confusion in the text and the figures with these words. Also, it is not clear what are the percentages that are presented in this figure. Are they relative abundances of OTUs?

In Figure 3, graphs B, D, and F are relative abundance but they are not given as a sum of 100 or 1. They seem absolute abundances.

Figure 4 shows well the relative abundances of phytoplankton and zooplankton. However, it would enable the comparison between the two periods if the authors were presenting together the results of the wet and dry period for each sample instead of presenting first the dry and then the wet period. The authors should consider present 01_Dry, 01_Wet, 02_Dry, 02_Wet, etc

Similarly, in Figure 7, where the authors test for differences in the relative abundances of zooplankton and phytoplankton taxa, the numbers in the axis of the box plots does not represent relative abundances. The authors should be more descriptive on the legend to explain what this figure represents.

In Figures 8 and 10, the authors should consider increase the size of the font to make the figure more readable.

The raw data are not publicly available yet. The authors only provide the zOTU sequences as Supplementary material, but they should consider providing the MiSeq sequencing results in a Figshare file to facilitate data reanalysis in the future.

Experimental design

This study aims to describe the plankton community of the Xiao Jiang River using environmental DNA sequencing and to set the basis for the future management of water pollution. However, it is not clear how the research presented here fills an identified knowledge gap. Has the biodiversity of this ecosystem been previously studied using morphology or is this the first attempt to characterise the plankton diversity study in the area? Are there any previous findings to support that the plankton community composition and structure differ between the wet and dry period or is this a hypothesis to confirm in the present study? The research topic seems to be relevant and timely but the research question should have been better defined. The knowledge gap being investigated should be identified in a more straightforward way, and statements should be made as to how the study contributes to filling that gap.

The materials and methods section should be described in greater detail in order for the results to be reproducible.

Line 178: “DNA extraction and amplification were conducted on an ultra-clean table”
I am not familiar with the term “ultra-clean table”. I think that what the authors mean here is that they performed their experiments in a previously well-cleaned bench. However, disinfecting and cleaning is a very standard lab procedure that is not important to be mentioned in the section of methods.

Line 184: “3µL of DNA template”
Were the DNA samples previously normalised to a standard concentration ( e.g. 100 ng/μL ) ?

Line 192: “MiSeq platform (Illumina) was used for computer sequencing”
Replace “computer sequencing” with “DNA sequencing”

Lines 194-195: “Double-ended sequence splicing was performed using the software PEAR”
Do the authors mean “pair-end read merge”? The software described here is a merger and I am not aware of a splicing step in the bioinformatic protocol of Illumina short reads.

Lines 200-202: “ taxa other than plankton were removed”
How did the authors defined plankton taxa? Did they chose those taxa based on previous literature? This is a very important detail for their results to be reproducible by other researchers, thus they should provide a more detailed explanation on data selection and OTUs filtering. The filtering rationale should be explained in greater detail. Which groups of organisms have been removed and which have been retained for downstream analysis and what were the criteria for doing so? The authors should add references to justify their filtering.

Lines 203-215: “All statistical analyses were done in R 4.0.5.”
The authors describe the statistical analyses but they do not provide the names and versions of the packages that they used for each analysis. The authors should be more explicit regarding this information, in order for their results to be reproducible.

Lines 208-210: “To evaluate the temporal variation of the plankton community, a principal coordinate analysis (PCoA) was performed for each sample by sampling period based on Bray-Curtis”
This sentence does not make sense as it is impossible to perform PCoA for each sample; this analysis is performed in a set of samples to evaluate their ordination. I suppose that the authors mean that they have calculated the Bray-Curtis distance for each sample and they performed a principal coordinate analysis (PCoA) based on this distance, colouring the samples as from the wet and the dry period.

The authors should consider rewrite the entire Materials and Methods section properly, taking care of the English language and including all the information needed for their research to be reproducible.

Validity of the findings

Lines 232-233: “the plankton species in the wet period nearly covered the species in the dry period”
It is not clear what the authors mean here. In which sense the species in the wet period covered the species in the dry? In terms of composition? In terms of richness? “Covered” is not a very informative verb to describe these results.

Lines 233-241: The results presented in these lines are confusing as the information is not very well-structured. First, it is not clear what these percentages are representing. Should the reader add them up to a total 100% somehow? Also, how is the species composition expressed as a percentage of anything?

Lines 246-248: “In summary, the species composition of zooplankton and phytoplankton was consistent in both the dry and wet periods”
Does this mean that the exact same species are present in the two periods? I think that the authors are describing the community at phylum level and they should be more explicit to avoid confusion.

Lines 259-260: “Overall, the Shannon and Simpson indices indicated that the wet period was greater than the dry period, but the Pielou index was significantly higher in the dry period than in the wet period.”
"The wet was greater than the dry period” in terms of what? I suppose the authors refer to species richness but this should be explicitly stated.

---

## Round 0.2 · Minor Revisions

Although reviewers have noted significant improvements in your revised manuscript there are minor comments and changes you still need to make. I need you to revise your manuscript by considering the comments raised by the reviewer.

·

Basic reporting

I would like to thank the authors for the improving the manuscript, but some concerns remain to be addressed, see next sections.

Experimental design

Please provide NCBI BioProject PRJNA941381 number. Check, that all SRA records are included into the BioProject. Now Genbank shows “No items found” for BioSample and SRA in PRJNA941381.
Figure 4. The indices have different scale, so using the same color palette is not correct, please use separate color bar for every index.

Validity of the findings

L331-332: Sentence is not clear, please rephrase.
L358: It’s not clear which percentage stands for what case, please rephrase.
Conclusion section must include all key finding of the study, e.g. species relationship or particular environment factors which have influence on microbiome.

---

## Round 0.3 · accepted · Accept

The authors have addressed all the comments raised during the review process, and the manuscript is acceptable for publication.